# Methylmercury plus Ethanol Exposure: How Much Does This Combination Affect Emotionality?

**DOI:** 10.3390/ijms222313131

**Published:** 2021-12-04

**Authors:** Diandra Araújo Luz, Sabrina de Carvalho Cartágenes, Cinthia Cristina Sousa de Menezes da Silveira, Bruno Gonçalves Pinheiro, Kissila Márvia Matias Machado Ferraro, Luanna de Melo Pereira Fernandes, Enéas Andrade Fontes-Júnior, Cristiane do Socorro Ferraz Maia

**Affiliations:** 1Laboratório de Farmacologia da Inflamação e do Comportamento, Instituto de Ciências da Saúde, Universidade Federal do Pará, Belém 66075-110, PA, Brazil; diandra.arluz@gmail.com (D.A.L.); sabrina_decarvalho@yahoo.com.br (S.d.C.C.); cinthia_bel@yahoo.com.br (C.C.S.d.M.d.S.); bgonsalvespinheiro@hotmail.com (B.G.P.); kiss.linha@hotmail.com (K.M.M.M.F.); efontes@ufpa.br (E.A.F.-J.); 2Departamento de Ciências Morfológicas e Fisiológicas, Centro das Ciências Biológicas e da Saúde (CCBS), Universidade Estadual do Pará, Belém 66095-100, PA, Brazil; luannafe@hotmail.com

**Keywords:** mercury, ethanol, toxicology, depression, anxiety, insomnia, pollutant, emotional disorders

## Abstract

Mercury is a heavy metal found in organic and inorganic forms that represents an important toxicant with impact on human health. Mercury can be released in the environment by natural phenoms (i.e., volcanic eruptions), industrial products, waste, or anthropogenic actions (i.e., mining activity). Evidence has pointed to mercury exposure inducing neurological damages related to emotional disturbance, such as anxiety, depression, and insomnia. The mechanisms that underlie these emotional disorders remain poorly understood, although an important role of glutamatergic pathways, alterations in HPA axis, and disturbance in activity of monoamines have been suggested. Ethanol (EtOH) is a psychoactive substance consumed worldwide that induces emotional alterations that have been strongly investigated, and shares common pathophysiological mechanisms with mercury. Concomitant mercury and EtOH intoxication occur in several regions of the world, specially by communities that consume seafood and fish as the principal product of nutrition (i.e., Amazon region). Such affront appears to be more deleterious in critical periods of life, such as the prenatal and adolescence period. Thus, this review aimed to discuss the cellular and behavioral changes displayed by the mercury plus EtOH exposure during adolescence, focused on emotional disorders, to answer the question of whether mercury plus EtOH exposure intensifies depression, anxiety, and insomnia observed by the toxicants in isolation.

## 1. Introduction

Exposure to environmental toxicants has impact on human health worldwide. Mercury is an environmental pollutant produced by natural phenoms, industrial and mining activity, as well as by deforestation, which is toxic to the human neurodevelopmental period [1]. In the Amazon region, natural sources and anthropogenic actions represent the major causes of mercury exposure, which can be found as organic and non-organic chemical forms, in the riverine populations [2,3,4]. The non-observed adverse effect level (NOAEL), declared by the World Health Organization (WHO), in mercury concentrations of 10 μg g^−1^ in human hair has been exceeded in contaminated population regions, such as mining areas [5]. Despite the WHO NOAEL recommendation, it fails to delimitate relevant factors as low levels of long-term exposure still occurring or the past acute intoxication still detected in hair [6]. Thus, the establishment of secure levels of mercury exposure is complex, as well as the design of intoxication paradigm in animal models, which could reflect human exposure. Furthermore, mercury distribution, toxicity, and metabolism depend on its chemical form, for which methylmercury (MeHg), an organic derivate, has been extensively studied due to its ability to cross the blood–brain barrier, reaching high levels in the central nervous system (CNS) [7]. Therefore, in vitro and in vivo studies have been essential to elucidate the toxicological mechanisms that underlie the symptoms exhibited by humans following mercury exposure. In experimental challenges, mercury has been administrated by inhalation, oral, or intraperitoneal routes, while in humans the consumption of contaminated foods (i.e., fishes and seafood) reflects the most common intoxication profile. Nonetheless, doses that elicit mercury blood and hair levels similar to those found in clinical studies, in addition to differentiation of acute and chronic exposure, consist of strategies to minimize limitations related to animal studies.

Neurotoxicity of mercury has been described through in vitro and in vivo studies. The occurrence of these undesired neurological effects depends on the level of exposure, by which paresthesia, dysarthria, progressive constriction of the visual field, hearing loss, sensory deficits, ataxia, tremor, deterioration of cognitive functions and paralysis have been reported [8,9,10,11]. Epidemiological and experimental studies have reported the vulnerability of the developing brain to environmental pollutants such as MeHg, the most toxic form of mercury, including a decrease in cell proliferation in the developing neural tube [7,8,9,10]. Several mechanisms have been suggested, including negative effects on neurotransmitter systems, induction of oxidative stress, microtubule disruption, and intracellular calcium homeostasis disturbance [12,13,14,15]. The fundamental toxicological mechanism related to the CNS damage appears to rely on the induction of an excessive amount of synaptic glutamate (i.e., inhibition of reuptake by astrocytes), as well as stimulation of neuronal release of this neurotransmitter, consequently leading to excitotoxicity and cell death [16,17,18]. In addition, mercury also disturbs the cellular redox homeostasis, with overproduction of the reactive oxygen species (ROS) and reduction of antioxidant systems activity [19,20,21,22].

Beyond the detrimental effects of this heavy metal, the concomitant exposure to other neurotoxicants may increase the potential damage on CNS. In this sense, we highlight that the ingestion of ethanol (EtOH), which the intake initiates early in adolescence, has been commonly observed in mining regions, and perhaps may synergically intensify the mercury damage [1]. EtOH directly reduces neuronal plasticity in the early life, impairing cognitive function, emotionality, and motor skills, among other CNS domains in adult life [23,24]. Adolescence is known as a developmental period marked for several changes, including mood, at which point the susceptibility to disorders such as depression, anxiety, and insomnia increases due to hormonal and neuronal functions that undergo an intense process of modifications [25,26]. Whereas EtOH effects on adolescence at several patterns of consumption have been extensively investigated, findings on mercury contamination in adolescence are scarce, as well as investigations related to mercury plus EtOH intoxication. In this sense, further studies are necessary to respond to the existing scientific gaps.

Thus, the purpose of this review was to gather the information available about mercury effects on emotionality described and propose the possible pathophysiological mechanisms involved. Thus, depression, anxiety, and insomnia that were reported in epidemiological surveys were chosen [5,27]. Posteriorly, we hypothesize whether mercury exposure associated to EtOH intake can synergically potentiate the damage caused by mercury *per se*, including the molecular mechanism involved in the toxicological behavior disturbance. Due to the importance of developmental alterations that undergo immature brain and few studies performed on this life stage, we have chosen to discuss the possible toxicological effects on the adolescence period. In the absence of mercury plus EtOH studies, findings related to isolated substances were discussed to propose the toxicological mechanism. Furthermore, due to the gap in the literature at this stage of development, some findings of different periods of life were also gathered to construct the hypothetical toxicological mechanism when opportune. Our effort relies on the collection of information that highlights the silent danger related to the simultaneous intoxication by mercury and EtOH on the critical phase of development. Available literature was searched on Pubmed, with the keywords: mercury and/or ethanol, behavioral disturbance, behavioral alterations, emotional disorders, adolescence, depression, anxiety, and insomnia.

## 2. Toxicokinetic Interaction

Robust literature has indicated that mercury exposure deposits in body systems during neurodevelopment with distinct differences among body tissues [1,28,29]. In CNS, our group has demonstrated that MeHg exhibits different tropism, for which cortical tissues appear to be a preferential target in detrimental to central structures in adolescent animals [1]. Previous studies already have pointed to such differences in MeHg affinity for CNS regions [30].

Of note, EtOH modify MeHg toxicokinetic in the simultaneous exposure [1,28,31,32]. A survey related to women who drank alcohol concomitantly to a regular consumption of fish/seafood during pregnancy support this interaction [31]. Grandjean and Weihe [31] detected low levels of mercury on the cord-blood of children whose mothers had consumed EtOH compared to those who were abstinent during pregnancy The authors attributed this mercury reduced level to the presence of protector compounds on fish (i.e., polyunsaturated fatty acids), since the elevated frequency of fish consumption in pregnancy has been correlated to a lower average of mercury on cord-blood [31]. However, in a previous study, the authors fail to show that an acute low dose of EtOH exposure (1.0 g/kg) interferes with mercury tissue content [29]. We hypothesize that such disaccord relies on the mercury plus EtOH challenge, which a single dose of both toxicants, dosage, and only 6 h as a cut-off for mercury content evaluation, were not sufficient to observe the scenario of kinetics interactions. In contrast, Tamashiro et al. [32] suggested that animals who intake higher doses of EtOH (5% and 10%) for 10 consecutive days present higher mercury deposits in body tissues, such as the brain, which has been contested by another experimental assay [33]. We believe these contrasting findings could be related to the period of exposure, for which the mature brain may elicit a different profile of MeHg interaction with CNS tissue components.

Our previous study in EtOH binge drinking paradigm in the adolescent model revealed that mercury content in hair and CNS (i.e., cerebellum, prefrontal, and motor cortex) were reduced in the concomitant exposure to EtOH [1,34]. The mechanisms involved in chronic EtOH-induced detoxication of MeHg in brain tissues have been poorly understood. MeHg is detoxified through S-conjugation with glutathione, forming an MeHg-SG product, which is excreted by a transporter protein [34]. We suggest an alternative based on the overproduction of hydrogen sulphide (H2S) by enzymatic systems, which exhibits higher affinity by MeHg, to form the inactive metabolite (MeHg)2S, presenting that MeHg binds to thiol groups in tissue protein [1,28,35,36,37]. Such enzymatic system is inducible by EtOH exposure, which may explain the reduced deposit of MeHg in brain regions and hair. Despite this toxicokinetic interaction appears to elicit a protection against mercury toxicological effects, our studies demonstrated that the reduced mercury deposit did not minimize the behavioral changes, on the contrary, the reduced mercury tissue levels displayed an equal index of functional disturbance, which we postulated as a synergistic toxicological mechanism [1,34].

## 3. Behavioral Alterations

### 3.1. Depression

Studies postulate the relationship between long-term effects of exposure to MeHg (0.5 mg/kg/day) and increased risk for depression [38,39]. Depression affects more than 350 million people worldwide and is a leading cause of disability [40,41]. Nowadays, such disorder represents a major public health problem, since it interferes with social functioning, educational attainment, economic output, and overall quality of life [42,43,44,45].

In experimental studies, depressive behavior induced by MeHg has been investigated. Our group has explored MeHg during the neurodevelopment as a predisposing factor for a depressive-like profile in rats [28]. Firstly, we investigated the prenatal exposure, which we found that an acute exposure of MeHg in pregnant rats (8 mg/kg, by gavage on the 15th day of pregnancy) induced depression-like behavior during offspring adulthood, associated with down-regulation of nitric oxide (NO) synthesis enzymes [28,46]. Such findings reveal a persistent behavioral and molecular alteration following acute intrauterine exposure. Since NO physiological levels activate important pathways of synaptic plasticity (i.e., Akt/CREB), and act as neuroprotector against NMDA-mediated neurotoxicity, reduction in its levels can impair neuronal function, which can, at least in part, explain the depressive-like behavior observed [47]. Posteriorly, we performed a chronic low dose of MeHg exposure during adolescence that also induced depressive-like features in rats [1]. Curiously, it appears that MeHg exposure induced depression-like behavior was not dependent of brain maturation, since chronic low doses of this metal exposure also displayed a long-term depression-like profile in adult subjects [48,49].

Essentially, the neurochemical basis of MeHg-induced depression-like behavior relies on disturbances in different neurotransmitter systems, mainly glutamate, in the early-exposure, followed by long-lasting changes in brain functioning [39,46,50]. In rats, 30 μM of MeHg modifies glutamatergic synaptic transmission on the hippocampus, through changes on the glutamatergic terminal axon regions, modifying synaptic potential response and resulting in irreversible depression of synaptic efficacy [51]. In the glutamatergic pathway, an increment of glutamate concentrations in the synaptic cleft occurs that results in hyperactivation of N-methyl D-aspartate (NMDA)-type glutamate receptors, leading to an increase of intracellular Na^+^ and Ca^2+^ levels, which has been associated with generation of ROS, as well as the triggering of important pathways involved in cell death induced by increased extracellular Ca^2+^ levels, disrupting glutamate and calcium (Ca^2+^) homeostasis in intracellular compartments, including mitochondria [7,52,53,54,55,56,57,58,59]. The augment of Ca^2+^ levels provoke activation of important vias involved in cell death [55]. Such alterations impact on the mitochondrial electron transport chain (ETC), which in vivo and in vitro studies have revealed that MeHg toxicity alters the complexes II and III of the mitochondrial ETC, eliciting depression of respiratory mechanisms and ATP production, and swelling of the mitochondrial matrix, which may contribute to the pathophysiology of depression [56,60]. In fact, alteration in Ca^2+^ homeostasis; direct toxic effects on mitochondria resulting in mitochondrial damage/dysfunction; and induction of oxidative stress consists of aggravating factor in fundamental brain areas related to mood disorders (i.e., hippocampus) [61], including depression elicited by MeHg exposure [56,60].

In addition, brain-derived neurotrophic factor (BDNF) secreted by astrocytes regulates synaptic plasticity and memory formation in the brain, stimulates release of neurotransmitters in presynaptic neurons, and enhances ion channels in postsynaptic neurons [62,63,64,65]. BDNF also provides neuroprotection on the hippocampal region against ischemic damage, favoring the increase of antioxidant enzymes [3,64,66]. Studies have demonstrated that BDNF undergoes down regulation by MeHg exposure following prenatal, neonatal, and adulthood exposure in rats [67,68,69,70]. Thus, it is reasonable to infer that MeHg predisposes to depression disorders by a plethora of molecular mechanisms. In support of this hypothesis, studies have linked the depression-like behavior associated to the reduction of different Bdnf transcripts levels on the frontal cortex and hippocampus following MeHg exposure [39,69,70].

Interestingly, these experimental findings have recently been reflected in epidemiological studies. Unfortunately, scarce clinical studies have assessed MeHg as a contributing factor in depression. An epidemiological survey with Amazon riverine inhabitants who claimed to present depression symptoms demonstrated a total mercury level in hair at levels below of 10 μg g^−1^ [27]. In turn, Arrifano et al. [5] did not find statistical differences upon depression symptoms between groups with higher (≥10 μg g^−1^) and lower (<10 μg g^−1^) amount of total hair mercury, which suggests that occurrence of this psychiatric disorder was not dependent of mercury level exposure. However, complementary studies with more participants are necessary to clarify this issue, characterizing fundamental issues, as if the exposure is acute or chronic, as a well as to establish a more concrete relationship between depressive symptoms and mercury exposure. Despite the above-mentioned studies have not been focused on adolescent individuals, we suggest that chronic MeHg exposure might occur from early life, due to reduced mobility among Amazon riverine communities, as observed in both clinical works. In this context, Costa-Júnior et al. [27] only included individuals that inhabit such regions for at least a year and aged from 13 years. Previously, depressive disorder has been found in adult individuals exposed to MeHg in the early life of residents of Minamata (Japan) [71,72]. In line with this, we highlight the need for research specifically addressing adolescents.

Robust literature has identified EtOH intake as a risk factor in depression [73,74]. Interestingly, the relationship between EtOH consumption and depression-like phenotype sex-dependently has been proposed, including epigenetic factors as predisposing factors for depressive-like behavior induced by parental EtOH exposure [75]. In this sense, we highlight those emotional disorders involving EtOH consumption depends on multiple factors such as dose, period of consumption, developmental stages, and sex [76,77]. Indeed, EtOH shares common pathological mechanisms with MeHg on the pathophysiologic theory of depression. The EtOH-induced blockade of NMDA receptor reduces the glutamate-NMDA pathway activation and BDNF levels, impairs several neurotransmitter systems, alters mitochondrial homeostasis, induces oxidative stress and neuroinflammation, among other toxicological mechanisms [76,77], which appears to share common damage pathways between both toxicants [1,51,52,53,54,55,56,57,58,59,67,68,69,70].

In line with this, our group has made efforts to assess the neurobehavioral consequences resultant of interaction of MeHg plus EtOH during two distinct life periods in rodents [1,13,28,46,78]. Firstly, our findings fail to indicate depressive-like behavior in offspring adult life following concomitant perinatal exposure to MeHg (8 mg/kg on the 15th day of pregnancy) plus EtOH (6.5 g/kg per day during pregnancy and lactation) [28]. Since both neurotoxicants elicit depressive-like effects, this result was unexpected. Thus, we suggest that absence of depression-like behavior may be a consequence of overproduction of H2S induced by heavy EtOH administration on an acute single dose of MeHg exposure, possibly reducing MeHg concentration on brain regions of mood regulation through toxicokinetic adaptations that ethanol possibly imposes on MeHg excretion, in a long-lasting period [35,36,37].

In 2018, our group explored the concomitant intoxication challenge, using a model of chronic exposure to low doses (CELDS) of MeHg (40 μg g^−1^/day for 5 weeks) plus binge drinking EtOH (3.0 g/kg/day, 3-days-per-week, for 5 weeks) during the adolescence period [1]. Due to obvious limitation of animal models, such experimental design mimics the standard of consumption of those subjects. In contrast with our perinatal intoxication model, even the reduction of total mercury tissue and hair levels by EtOH interaction, depressive-like behavior was equally present in the forced swimming task, which suggests a potentiated toxicological effect [1]. Numerous reasons might be raised for such controversial data, such as the intoxication protocol (acute vs. chronic), window of neurodevelopment (perinatal vs. adolescence), and synergistical effects. On the other hand, we could deduce that chronic exposure, even at low doses during adolescence, appears to be more susceptible to mercury-induced alterations than acute prenatal intoxication. However, further studies need to be designed not only to confirm this finding, but to explain the reasons for this apparent vulnerability in this period of life.

Finally, we hypothesize that the toxicological mechanisms shared by both neurotoxicants on depression pathophysiological mechanisms might depend on the dose and life period exposure, as a well as the long-term cumulative intoxication of both neurotoxicants to provoke depression.

### 3.2. Anxiety

Adolescence is a developmental period for which major CNS structural and functional changes occur, particularly pruning and maturation of the prefrontal region that participates in the limbic system, as well as the neuroanatomical circuitry of anxiety [79]. Curiously, the prefrontal cortex and hippocampus that undergo maturation during adolescence also have been described as targets of the toxic effects of mercury, consisting of two fundamental structures for emotional behavior [1,13].

Belém-Filho and colleagues [1] observed increased mercury content levels in the prefrontal region compared to other brain areas associated to anxiogenic-like behavior. Furthermore, reduced GABAergic signaling sensibility on the prefrontal cortex supports the mercury hazardous effects related to anxiety [80].

In vivo experimental analyses have reported pathophysiological changes on the hippocampus after perinatal and postnatal exposure to MeHg by which the neurotoxic effects at cellular level have been correlated with oxidative stress, excitotoxicity, damage to deoxyribonucleic acid (DNA), neurogenesis alterations, calcium (Ca^2+^) homeostasis disruption, neuroinflammation, and cell death mechanisms [13].

Long-term anxiety-like behavior was observed in rats prenatally exposed to acute high doses of MeHg [28,46]. Interestingly, an anxiety-like profile has also been seen in our model of CELD of MeHg during adolescence [1]. In addition to the prefrontal cortex, multiple important signaling cascades on the hippocampus may underlie behavioral deficits, such as anxiety. Disorders in fear neurocircuitry have resulted from the hippocampus-prefrontal cortex signaling, reflecting a clinical manifestation of anxiety [79]. Thus, evidence of hippocampal tissue alteration induced by MeHg has been demonstrated [1,79]. In a pregnant rat, an MeHg exposure model (1.0 or 2.0 mg/kg from gestational day 5–21) resulted in adolescent pups exhibiting an anxiety-like phenotype associated with down-regulation of the PI3K/Akt/mTOR pathway, disrupting cell growth, differentiation, and survival, as well as triggering apoptotic events [81]. In addition, disruption on neuroplasticity through reduction of GSK3β activity has been identified in the neurodeveloping hippocampus [81]. Cytoskeletal protein homeostasis dysregulation resulted by altered MAPK/NFM-NFH signaling, as well as the up-regulation of pro-apoptotic pathway (i.e., BAD/BCL-2, BAX/BCL-XL, and caspase reinforces the vulnerability of neuronal function to MeHg [81]. However, it is necessary to investigate if these pathophysiologic mechanisms described above during prenatal exposure reflects those that occur during adolescence intoxication.

As expected, glutamatergic neurotransmission impairment was also evidenced in an acute MeHg exposure during pregnancy (8 mg/kg, on gestational-days 8 or 15) [82]. This glutamate signaling negative effects has been related to a lower level of exploratory behavior, as well as a deficiency in habituation behavior on the new object exploration task [82].

Furthermore, in cortical cell cultures from animals exposed to MeHg during pregnancy, basal levels of extracellular glutamate were higher, while levels of extracellular glutamate evoked-KCl were lower than control sample [82].

In addition, the hypothalamic–pituitary–adrenal (HPA) axis function has been affected by MeHg, which may result in anxiety disorders [83]. In other words, anxiety induced by MeHg during the neurodevelopmental stages shares some molecular pathways observed in depression pathophysiological mechanisms described anteriorly. We highlight the importance of further evaluations during the adolescence period, due to specific changes displayed on the CNS at this period of life.

In humans, a prospective pregnancy/birth cohort investigation reported evidence of an association between exposure to very low doses of mercury during early pregnancy and anxiety scores in children [84]. Important epidemiological studies have actually linked exposure to MeHg to psychiatry disorders, as anxiety [5,27].

Considering that the brain’s structural maturation persists during adolescence, the exposure to EtOH in early life may produce neurobehavioral consequences. Our group has explored chronic EtOH intake patterns during perinatal and adolescence life, mainly by the heavy or binge drinking paradigm in rats [1,23,24,85]. All these EtOH challenges indicated persistent anxiogenic responses, associated to reduction in the volume of hippocampal formation, alteration of nitrergic activity, as well as cellular disturbance (i.e., neurons, astrocytes, and microglia) on the hippocampus [1,23,24,46,75,85]. In addition, BDNF level reduction and astrocyte activation have been described [76]. These changes in cellular homeostasis may trigger neuronal apoptosis [76,77,85], as described above [7,21,51,52,53,54,55,56,57,58,59,67,68,69,70]. It is relevant to note that the severity of alcohol toxicity effects is related to drinking pattern and time of exposure, in which intermittent alcohol exposure affects increased alcohol intake during the pre-gestational period and lactation differently than the continuous access [86].

In contrast, a study has not observed anxiety-like responses in an elevated plus-maze test after intermittent moderate alcohol consumption by male and female adolescent animals. Nonetheless, a decrease in several markers of dopaminergic and serotoninergic neurotransmission has been described [87]. Serotonin (5-HT) represents a fundamental neurotransmitter contributing to the CNS organization and related to psychiatric disorders, particularly anxiety [88]. Curiously, Bellia et al. [88] has postulated that transient 5-HT depletion during adolescence reduces anxiety-like behavior in female rats, which was accompanied by a reduction in EtOH intake. Such fact suggests that serotonin plays an important role on EtOH addiction and anxiety. Acetaldehyde, the first product of EtOH metabolism and one of the mediators of its peripheral and central effects, also presents rewarding, motivational, and additive properties due to its interaction with dopaminergic and endocannabinoid systems [89]. Such findings highlight the multiple targets elicited by EtOH and its metabolites on CNS, altering mood and generally increasing behavioral reactivity [90]. All these findings illustrate the pleiotropic effects of EtOH in the pathophysiology of anxiety, in which the exploration of each toxicological mechanism might be applied on adolescence.

Although studies of the MeHg plus EtOH exposure effects during adolescence are scarce, there is evidence of an increase in anxiety-like behavior after MeHg administration and the potentiation of this effect in the concomitant exposure with EtOH [1,28,46]. In work conducted by our group, it was found that offspring exposed to EtOH (6.5 g/kg per day, by gavage during pregnancy and lactation) plus MeHg (8 mg/kg, by gavage on the 15th day of pregnancy) during intrauterine life, demonstrated panic-related behavior [46], corroborating a previous study, which has suggested that intoxication by EtOH and/or high doses of MeHg during CNS development may be a risk for anxiety-related behavior [28].

Our group has also demonstrated that exposure to low doses of MeHg and/or associated to EtOH in adolescent rats elicited an anxiety-like phenotype, as well as peripheral oxidative stress [1]. In addition, mercury tropism for cortical structures and low affinity for central structures have been reported [1]. This augmented concentration of total mercury on the prefrontal area has been related to anxiogenic-type behavior, which reduced risk assessment, a parameter associated with anxiety disorders, was found in MeHg plus binge drinking EtOH treatment individuals [1]. These data are consistent with Maia et al. [46] who also found a reduction in risk assessment parameter in animals submitted perinatally to heavy chronic EtOH exposure plus acute MeHg intoxication. Chauhan and Chauhan [91] investigating exposure to MeHg in *Drosophila melanogaster* (fruit fly), observed the generation of free radicals and lipid peroxidation (markers of oxidative stress) in a dose-dependent manner, an effect potentiated by the association with EtOH. Furthermore, EtOH plus MeHg exposure did not affect the immobilization of flies but increased the recovery time in a concentration-dependent manner, suggesting that MeHg inhibits alcohol dehydrogenase activity in a concentration-dependent manner, impairing the elimination of the EtOH in the body system, reflecting the toxicokinetic interaction, already described above.

Together, as described in the depression section, it is reasonable to suggest that the exposure to mercury plus EtOH synergically induces negative effects on anxiety-like pathophysiology, since mercury levels in EtOH exposure subjects was reduced, though exhibiting equal effects [1]. Moreover, nitrergic activity enhancement and lipid peroxidation was also potentiated by MeHg plus EtOH exposure [1,78,91]. Complementary studies are required to clarify the real impact and pathological mechanisms related to mercury plus EtOH intoxication on anxiety, especially during adolescence.

### 3.3. Insomnia

Although insomnia has not been considered an emotional disorder, this disturbance commonly integrates the symptomatology of depression, anxiety, and other psychiatry disorders. Costa Junior et al. [27] conducted epidemiological research focused on emotional and motor manifestations in Amazonian riverine populations exposed to mercury. Residents from Itaituba City that presented higher levels of mercury than those of the Acará City population, have suffered insomnia isolated or associated to emotional disturbances. Posteriorly, Arrifano et al. [5] has accessed CNS disorders in riverine communities chronically exposed to MeHg. In 41.9% of patients presented with high levels (≥10 μg g^−1^) and 47.7% low levels (≤10 μg g^−1^) of mercury in hair and exhibited insomnia. In this research, alcohol drinking consisted of an exclusion criterion, which permits to deduce that this response is not related to EtOH intake. Although these two studies have not detected a direct correlation, such symptoms were recorded, highlighting the importance of identifying sleep biomarkers in MeHg intoxication.

In these studies, the population surveyed presented an average of 18–40 years old and were inhabited in the region for a long period, which suggests the probable long-term exposure to mercury, including adolescence. Besides, as commented previously, Costa-Júnior et al. [27] have included individuals that inhabit the region for at least one year from the interview. In addition, 13-year-old subjects have also been included in this work. However, it is not possible to know whether the insomnia reported by riverside dwellers was manifested from adolescence until adulthood or whether it was a cumulative clinical outcome.

Sleep cycle is regulated by environmental conditions (i.e., luminosity), hormonal signals, and neurotransmitter release. Melatonin is the main hormone reported as regulator of dark/light cycle, whereas other biomarkers, as cortisol, can also influence the initiation and duration of the sleep process [92]. Neurotransmitters also have been directly involved in induction and transition phases of sleep, in which the adequate function of all neurotransmitter systems, as well as other dependent processes may impact on cognition, general organic recovery, emotionality, etc. [92]. In a canonical pathway, reduction of luminosity, levels of substances on blood and brain (glucose, cortisol, adrenaline, among others), release of melatonin, followed by increases in GABAergic and galanin transmission results in the end of the wake phase [92,93,94]. On the other hand, imbalance on activity of other regulators such as serotonin, noradrenalin, dopamine, histamine, and glutamate also may result in sleep disturbance [94,95].

In fact, there are two process of sleep regulation called homeostatic and circadian, in which the former regulates the sleep length and depth, whereas the later regulates patterns of maximum sleepiness and maximum alertness throughout the 24 h day [96,97].

Some studies have claimed that such altered homeostatic sleep system induces mood disorders (increase of emotional reactivity/impulsivity), decreasing the connectivity between the prefrontal cortex and amygdala, and consequently increasing sleep deprivation, affecting numerous executive functions such as attention, working memory, inhibition, and cognitive flexibility [98].

Interactions between neurotransmitter systems that affect circadian cycle markers, such as melatonin (one of the sleep cycle promoters), can predispose to sleep disorders. In mammalians, noradrenergic inputs of the suprachiasmatic nucleus (integrative part of hypothalamus) to the pineal gland results in expression of serotonin N-acetyltransferase, an enzyme that converts serotonin in N-acetylserotonin, which is posteriorly methylated by hydroxyindole-O-methytransferase generating melatonin [99]. Such noradrenaline released by the suprachiasmatic nuclei promotes melatonin synthesis during the night and the expression of serotonin N-acetyltransferase are negatively regulated by activation of the glutamatergic system [100]. Considering that mercury induces hyperactivation of glutamatergic pathway, it is reasonable to conclude the involvement of glutamate on sleep disturbance is provoked by mercury intoxication, which deserves further investigation. Furthermore, animal studies revealed that chronic MeHg exposure elicits up-regulation of acetylcholine receptor and acetylcholinesterase inhibition [101], allowing the activation of dopamine projections involved in the cortico-striato-pallidal via, also regulated by glutamatergic projections, affecting the sleep–wake state [102].

Besides this possible pathophysiological mechanism, the HPA axis plays a pivotal role on sleep regulation, where traditionally deep sleep induces HPA inhibition [103]. On the other hand, HPA activation can trigger and exacerbate an endocrine response by corticotropin releasing hormone (CRH), followed by adrenocorticotropic hormone (ACTH) secretion from anterior hypophysis, stimulating the secretion of cortisol blood levels by the adrenal cortex [103]. Such cortisol stimulation triggers a stress situation that can induce insomnia and other psychiatric symptoms. Although the knowledge that mercury chloride, MeHg, or mercury vapor can form deposits in the hypothalamus of rats, the impact of this metal deposit in the sleep cycle is unclear [104,105]. An experimental study has postulated that citrate mercuric administration promotes changes in the circadian rhythm by reduction of cytoplasmic RNA of neurons in the hypothalamus, reflecting in hypothalamic dysfunction [106]. In this context, several possible conditions may be involved: (1) mercury accumulation in the pineal gland decreases melatonin and serotonin secretion [107]; (2) low melatonin levels decrease protective mechanisms important in the detoxification of mercury, as well as selenium and glutathione [108]; (3) these serotonin and melatonin reductions induce adaptive responses to free radicals formed, as overall production of NO, mainly on the suprachiasmatic nucleus (integrative part of the HPA axis), which affects operate capacity of the supra-chiasmatic nucleus in a similar manner to a light signal stimulus [96]; (4) such possible abnormalities may result in endogenous desynchronies in the light/dark cycle, as cortisol levels, sleep/wakefulness, and temperature [109]. Besides, mercury can deposit on the pituitary gland in animals and humans, principally by intracellular changes [110,111]. All of these hypothesized dysfunctions may impact on sleep alterations and for this reason, deserves further well-designed investigations.

Possible damage of the HPA axis or stress situation reinforces the idea of sleep disorders that result in release of these hormones, as observed in studies of single administration of MeHg in rats. One hour following subcutaneous 12 mg/kg of MeHg, elevated ACTH levels with reversible peak in the serum have been observed [112]. Moreover, different forms of mercury compounds can deposit on the adrenal gland in the cortex and medulla regions, interfering with adrenal maturation and increasing corticosterone release, affecting HPA axis [113].

In turn, EtOH intake seems to inhibit melatonin secretion dose-dependently, probably by diminution of tryptophan hydroxylase activity and phase delay in arylalkylamine N-acetyltransferase gene expression, beyond impairment in signalization of noradrenaline in the pineal glandule, with reduction of mRNA expression of β1 and α1 adrenergic receptors [114,115]. Additionally, EtOH exposure induces neurological adaptations, especially in conditions of alcoholism, in which up-regulation in glutamatergic function and down-regulation in GABA signaling result in symptoms related to agitation and increased alertness [116,117].

In the HPA axis, EtOH elicits HPA axis disturbances, mainly during withdrawal syndrome. Primarily, EtOH alters CRH releasing, impairing ACTH levels [117,118]. Indeed, EtOH induces endocrine modifications in the hormonal secretion in the adrenal gland with increases of cortisol levels [118,119]. Such toxicological mechanisms directly alter the stimulation of the pituitary and glucocorticoid secretion, as well as indirectly-by its metabolite acetaldehyde-, in an ACTH-independent manner, acting in adrenocortical cells [120]. Studies in humans have postulated that negative effects after EtOH heavy drinking paradigm were measured by diurnal changes in plasma cortisol, resulting in high concentrations during acute withdrawal [121]. Together, these present clinical implications in the sleep cycle of circadian rhythms. Mathematical hypothesis reinforces that acute/chronic EtOH exposure augments cortisol levels and amplitude of ultradian cortisol rhythm dose-dependently, consequently impairing HPA axis homeostasis [122].

Thus, considering the mechanisms on HPA axis and endocrine alterations, the association between mercury and EtOH may potentiate the risk of insomnia development. As mentioned, mercury and EtOH exposure increases glutamate activity, in which such shared toxicological mechanism can aggravate this condition. Until the present moment, data about the relationship between both toxicants during neurodevelopment remains unknown, reinforcing the urgent necessity of investigations in this field.

All hypothetical pathophysiological mechanisms related to MeHg and/or EtOH exposure on depression, anxiety, and insomnia are showed in Figure 1.

## 4. Conclusions

Despite mercury appearing as a widely dispersed environmental pollutant with severe neurotoxic effects on neurodevelopment, this metal occurs at high levels in the environment contaminated by anthropogenic activities (i.e., industrial waste, mining activity, and deforestation); however, few studies have investigated its effects on emotionality in childhood and adolescence period. Such lack of knowledge becomes worse in the simultaneous exposure to other neurotoxicants widely consumed during adolescence, such as alcohol. The present review highlights, for the first time, that mercury exposure during neurodevelopment elicits hazardous effects on emotionality disorders, such as anxiety, depression, and insomnia. Furthermore, in a unique review work, we discuss the synergically hazardous effects related to mercury plus EtOH exposure in adolescence. Finally, we emphasize common target mechanisms in both neurotoxicants (i.e., glutamatergic, dopaminergic, and serotonergic pathways; oxidative stress; and neuroinflammation), which reinforces the findings of the few experimental studies that have revealed the synergistically toxicological effects on depression, anxiety, and insomnia.

## Figures and Tables

**Figure 1 ijms-22-13131-f001:**
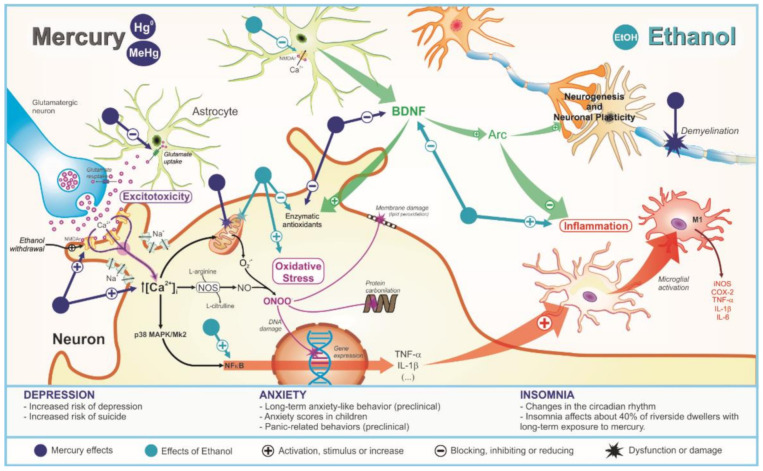
Pathophysiological mechanisms related to deleterious effects of mercury exposure on sleep and emotionality (depression and anxiety) disturbances and its potential synergism with ethanol consumption.

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
