# Peer review of "Methylmercury plus Ethanol Exposure: How Much Does This Combination Affect Emotionality?"

_ijms, 2021, doi:10.3390/ijms222313131_

Round 1

Reviewer 1 Report

THE AIM OF THE REVIEW IS TO SUMMARIZE THE YAT UNCLEAR EVIDENCE ON THE SYNERGISTIC EFFECT OF METHYLMERCURY AND ETHANOL CO-EXPOSURE.

THE STRUCTURE OF THE MANUSCRIPT IS WELL ORGANIZED.

HOWEVER, MAJOR REVISION IS MANDATORY, SINCE IT LOOKS LIKE A JUMBLED AND NOT-CONSEQUENTIAL DRAFT.

ENGLISH WRITING, PUNCTUATION AND GRAMMAR NEED AMENDMENTS, ALL OVER THE TEXT.

LINES 50-51: DISORDER IS NOT CORRECT SINCE THE AUTHORS (AA) REFER TO CNS DAMAGE

LINES 52-53: WHAT IS THE MEANING OF NEURONAL PRODUCTION? NEUROGENESIS?

LINE 90: THE AUTHORS SHOULD DEEPEN DETAILS OF THE SO CALLED “CO-INTOXICATION” PARADIGM

LINE 95: EXPAND ON “HIGHER DOSES”

LINE 100: THE INTERACTION IN PK BETWEEN ETOH AND ME-HG IS OBSCURE. AA NEED TO DESCRIBE IT CLEARLY EVEN IF A PUTATIVE MECHANISM IS PROPOSED

LINE 106: THE SENTENCE AT ROW 106 PAG 3 SUGGESTS THAT ETOH PREOTECTS FROM ME-HG INDUCED DAMAGE. IS THIS WHAT THE AA MEAN?

LINE 118: EXPAND ON NEUROTOXIC MECHANISM OF DOWNREGULATION OF NO SYNTHESIS

LINE 123: EXPAND ON THE NEUROCHEMICAL UNDERPINNING AND THE BEHAVIOURAL OUCOMES

LINES 136-138: NMDA NEUROTOXICITY INVOLVES NOS INDUCTION. CAN YOU EXPLAIN THE CONTRADDICTION WITH WHAT IS SAID BEFORE LINE 123-124?

LINE 157: WHAT KIND OF EVIDENCE SUPPORT THE SUGGESTION BESIDES THE PERMANENT RESIDENCY OF THE INHABITANTS OF AMAZONIA?

LINES 167-8: UNCLEAR SENTENCE

LINES 173-9: INSERT REFERENCE ON THE SUPPOSED MECHANISM OF CENTRAL TOXICITY OF ETOH

LINE 188: WHAT ARE THE EVIDENCE OF A ME-HG INDUCED DAMAGE IN THE PFC?

LINES 199-200: HOW ARE THE MULTIPLE SIGNALING CASCADE ALTERATIONS RELATED SPECIFICALLY TO ANXIETY DISORDERS?

Line 230: worthy of mention is Brancato, A., Plescia, F., Lavanco, G., Cavallaro, A., Cannizzaro, C.Continuous and intermittent alcohol free-choice from pre-gestational time to lactation: Focus on drinking trajectories and maternal behavior(2016) Frontiers in Behavioral Neuroscience, 10 (MAR), art. no. 31, 11

Lines 264-7: also acetaldehyde has risen concern in its ability to alter emotional, reactivity, learning and memory. see for reference

Plescia, F., Brancato, A., Marino, R.A.M., Vita, C., Navarra, M., Cannizzaro, C. Effect of acetaldehyde intoxication and withdrawal on NPY expression: Focus on endocannabinoidergic system involvement (2014) Frontiers in Psychiatry, 5 (OCT), art. no. 138

Cacace, S., Plescia, F., La Barbera, M., Cannizzaro, C. Evaluation of chronic alcohol self-administration by a 3-bottle choice paradigm in adult male rats. Effects on behavioural reactivity, spatial learning and reference memory (2011) Behavioural Brain Research, 219 (2), pp. 213-220.

LINE 290: DOES THIS RESEARCH DETECT DIRECT CORRELATION?

LINES 298-301:  THE AUTHORS MAY INTEND TO DEEPEN THIS PARAGRAPH??

LINES 302-7: SENTENCES LACK CORRELATION OR LINK BETWEEN CAUSE AND EFFECT. IN THIS CASE, BETWEEN NEUROTRANSMITTERS/ SLEEP/ MERCURY. AUTHORS SHOULD FORMULATE MORE FOCUSED SENTENCES WITHOUT DIGRESSIONS.

LINE 313: SENTENCE NOT ENTIRELY CORRECT, PLEASE SPECIFY PATHOLOGICAL CONDITIONS THAT LEAD TO THE EXACERBATED ENDOCRINE RESPONSE INVOLVING HPA AXIS

Author Response

October 25, 2021

MDPI IJMS Editorial Office

Manuscript: ijms-1403512

Title: "Methylmercury plus ethanol exposure: how much this combination affects

emotionality?"

Dear Editor,

We first wish to thank you for the opportunity to allow a resubmission of a revised and now ameliorated manuscript. We extend our thanks to the Reviewers for their positive evaluation of our study and for their generous suggestions to clarify the manuscript: their time and efforts were very warmly appreciated. We addressed all their criticisms and, guided by the suggestions, we revised the manuscript accordingly. The detailed answers to each of the questions raised are tendered point-by-point, in order of appearance, as follows:

#Reviewer 1

Lines 50-51: disorder is not correct since the authors (aa) refer to CNS damage

Response: Thanks. The term was substituted for “damage” on sentence as required. (line: 66-67, page 02)

Lines 52-53: what is the meaning of neuronal production? Neurogenesis?

Response: The sentence was adjusted for a better comprehension: “The fundamental toxicological mechanism related to the CNS damage appears to rely on the induction of an excessive amount of synaptic glutamate (i.e., inhibition of reuptake by astrocytes), as well as stimulation of neuronal release of this neurotransmitter, consequently leading to excitotoxicity and cell death [16-18].” (line: 66-70, page 02).

Line 90: the authors should deepen details of the so called “co-intoxication” paradigm

Response: We are very grateful for your recommendation and guidance, which we followed replacing the previous reference by “Of note, EtOH modify MeHg toxicokinetic in the simultaneous exposure [1, 28, 31, 32].” (line: 110-111, page 03).

Line 95: expand on “higher doses”

Response: We thank the Reviewer for noting this omission; We have clarified our sentence, as follow: “However, in a previous study, the authors fail to show that acute low dose of EtOH exposure (1.0 g/kg) interferes with mercury tissues content [29]. We hypothesize that such disaccord relies on the mercury plus EtOH challenge, which a single dose of both toxicants, dosage, and only 6h as a cut-off for mercury content evaluation were not sufficient to observe the scenario of kinetics interactions. In contrast, Tamashiro et al [32] suggested that animals which intake higher doses of EtOH (5% and 10%) for 10 consecutive days present higher mercury deposit in body tissues, as brain, which has been contested by another experimental assay [33].” (line: 117-124, page 124).

Line 100: the interaction in pk between EtOH and me-hg is obscure. Aa need to describe it clearly even if a putative mechanism is proposed

Response: We thank the Reviewer for these suggested corrections, which really helped to make the presentation clearer. We have introduced the following findings and hypothesis, as follow: “Our previous study in EtOH binge drinking paradigm in adolescent model revealed that mercury content in hair and CNS (i.e., cerebellum, prefrontal, and motor cortex) were reduced in the concomitant exposure to EtOH [1, 34]. The mechanisms involved in chronic EtOH-induced detoxication of MeHg in brain tissues have been poorly understood. MeHg is detoxified through S-conjugation with glutathione, forming MeHg-SG product which is excreted by a transporter protein [34]. We suggest an alternative via based on the overproduction of hydrogen sulphide (H2S) by enzymatic systems, which exhibits higher affinity by MeHg, to form the inactive metabolite (MeHg)2S, preventing that MeHg binds to thiol groups in tissue protein [1, 28, 35, 36].  Such enzymatic system is inducible by EtOH exposure, which may explain the reduced deposit of MeHg in brain regions and hair. Despite this toxicokinetic interaction appears to elicit a protection against mercury toxicological effects, our studies demonstrated that the reduced mercury deposit did not minimize the behavioral changes, on the contrary, the reduced mercury tissue levels displayed an equal index of functional disturbance, which we postulated as a synergistic toxicological mechanism [1, 34].” (line: 127-141, page 03).

Line 106: the sentence at row 106 pag 3 suggests that EtOH protects from me-hg induced damage. Is this what the aa mean?

Response: We thank the Reviewer for noting this aspect. The EtOH was not able to protect against mercury damage, solely by reducing tissue deposits. We now clarify this issue, as follow: “Such enzymatic system is inducible by EtOH exposure, which may explain the reduced deposit of MeHg in brain regions and hair. Despite this toxicokinetic interaction appears to elicit a protection against mercury toxicological effects, our studies demonstrated that the reduced mercury deposit did not minimize the behavioral changes, on the contrary, the reduced mercury tissue levels displayed an equal index of functional disturbance, which we postulated as a synergistic toxicological mechanism [1, 34].” (line: 135-141, page 03).

Line 118: expand on neurotoxic mechanism of downregulation of NO synthesis

Response: Following the Reviewer’s suggestion, we have expanded this issue, as follow: “Since NO physiological levels activates important pathways of synaptic plasticity (i.e., Akt/CREB), and acts as neuroprotector against NMDA-mediated neurotoxicity, reduction in its levels can impair neuronal function, which can, at least in part, explain the depressive-like behavior observed [47].” (line: 157-160, page 04).

Line 123: expand on the neurochemical underpinning and the behavioural outcomes

Response: We thank the Reviewer for noting this poorly explained issue. We now added fundamental information, as follow: “In glutamatergic pathway, occurs an increment of glutamate concentrations in the synaptic cleft that results in hyperactivation of N-methyl D-aspartate (NMDA)-type glutamate receptors, leading to an increase of intracellular Na+ and Ca2+ levels, which has been associated with generation of ROS, as well as triggering of important pathways involved in cell death induced by increased extracellular Ca2+ levels, disrupting gluta-mate and calcium (Ca2+) homeostasis in intracellular compartments, including mitochondria [7, 52, 53, 54, 55, 56, 57, 58, 59]. The augment of Ca2+ levels provoke activation of important vias involved in cell death [55]. Such alterations impact on the mitochondrial electron transport chain (ETC), which in vivo and in vitro studies have revealed that MeHg toxicity alters the complexes II and III of the mitochondrial ETC, eliciting de-pression of respiratory mechanisms and ATP production, and swelling of the mitochondrial matrix, which may contribute to the pathophysiology of depression [56, 60]. In fact, alteration in Ca2+ homeostasis; direct toxic effects on mitochondria resulting in mitochondrial damage/dysfunction; and induction of oxidative stress consists of aggravating factor in fundamental brain areas related to mood disorders (i.e., hippocampus) [61], including depression elicited by MeHg exposure [56, 60]. (line: 170-186, page 04)

Lines 136-138: NMDA neurotoxicity involves NO induction. Can you explain the contraddiction with what is said before line 123-124?

Response: We very much thank the Reviewer for requesting this clarification. This was important to re-structure the revised manuscript. The sentence anteriorly in the lines 123 and 124 were adjusted for a better comprehension, which now includes: “Since NO physiological levels activates important pathways of synaptic plasticity (i.e., Akt/CREB), and acts as neuroprotector against NMDA-mediated neurotoxicity, reduction in its levels can impair neuronal function, which can, at least in part, explain the depressive-like behavior observed [47].” (line: 157-160, page 04).

Line 157: what kind of evidence support the suggestion besides the permanent residency of the inhabitants of Amazonia?

Response: We agree with the Reviewer that we have not provided data to support such a conclusion. However, we still believe that such possible existence, which we raised the hypothesis. To make this point clearer, we reformulated the sentence, as follow at depression section: “However, complementary studies with more participants are necessary to clarify this issue, characterizing fundamental issues, as if the exposure is acute or chronic, as a well as to establish a more concrete relationship between depressive symptoms and mercury exposure. Despite the above-mentioned studies have been not focused on adolescent individuals, we suggest that chronic MeHg exposure might occur since the early life, due to reduced mobility among Amazon riverine communities, as observed in both clinical works. In this context, Costa-Júnior et al. [27] only included individuals that inhabit such regions for at least a year and aged from 13 years. Previously, depressive disorder has been found in adult individuals exposed to MeHg early life in residents of Minamata (Japan) [71, 72]. In line with this, we highlight the need for research specifically addressing the adolescent people.” (line: 205-215, page 05). Besides, we included in the insomnia section: “In these studies, the population surveyed presented an average of 18-40 years old and inhabited in the region for a long period, which suggests the probable long-term exposure to mercury, including adolescence. Besides, as commented previously, Costa-Júnior et al. [27] have included individuals that inhabit at the region for at least one year from the interview. In addition, 13-years-of-age subjects have been also admitted in this work. However, it is not possible to know if the insomnia reported by riverside dwellers was manifested since adolescence until adulthood or whether it is a cumulative clinical outcome.” (line: 382-389, page 08).

Lines 167-8: unclear sentence

Response: We very much thank the Reviewer for requesting this clarification. We modified the sentence, as follow: “Since both neurotoxicants elicit depressive-like effects, this result was unexpectable. So, we suggest that absence of depression-like behavior may be a consequence of over-production of H2S induced by heavy EtOH administration on an acute single dose of MeHg exposure, possibly reducing MeHg concentration on brain regions of mood regulation through toxicokinetic adaptations that ethanol possibly imposes on MeHg excretion, in a long-lasting period [35-37].” (line: 233-238, page 05).

Lines 173-9: insert reference on the supposed mechanism of central toxicity of EtOH

Response: As requested, we included the references, as follow: “EtOH-induced blockade of NMDA receptor reduces glutamate-NMDA pathway activation and BDNF levels; impair several neurotransmitter systems; alters mitochondrial homeostasis; induces oxidative stress and neuroinflammation; among other toxicological mechanisms [76, 77], which appears to share common damage pathways between both toxicants [21, 51, 52, 53, 54, 55, 56, 57, 58, 59, 67, 68, 69, 70].” (line: 223-227, page 05).

Line 188: what are the evidence of a me-hg induced damage in the PFC?

Response: We thank the Reviewer for these suggested corrections, with which we fully agree. We reformulated the sentence, as follows: “Curiously, the prefrontal cortex and hippocampus that undergo maturation during adolescence also has been described as targets of the toxic effects of mercury, consisting of two fundamental structures for emotional behavior [1,13]. Belém-Filho and colleagues [1] observed increased mercury content levels in prefrontal region than other brain areas associated to anxiogenic-like behavior. Furthermore, reduced GABAergic signaling sensibility on prefrontal cortex supports the mercury hazardous effects related to anxiety [80].” (line: 260-266, page 06).

Lines 199-200: how are the multiple signaling cascade alterations related specifically to anxiety disorders?

Response: We are grateful for the Reviewer’s suggestions, which really helped to make the presentation clearer. We corrected the sentence kindly indicated by the Reviewer and made an effort (hopefully sufficient) to ameliorate the issue, as follows: “Actually, disorders in fear neurocircuitry resulted from hippocampus-prefrontal cortex signaling reflects a clinical manifestation of anxiety [79]. Thus, evidence of hippocampal tissue alteration induced by MeHg has been demonstrated [1, 79]. In a pregnant rat MeHg exposure model (1.0 or 2.0 mg/kg from gestational day 5-21), adolescent pups exhibited anxiety-like phenotype associated to downregulation of the PI3K/Akt/mTOR pathway, disrupting cell growth, differentiation, and survival, as well as triggering apoptotic events [81]. In addition, disruption on neuroplasticity through reduction of GSK3β activity has been identified in neurodeveloping hippocampus [81]. Cytoskeletal proteins homeostasis dysregulation resulted by altered MAPK/ NFM-NFH signaling, as well as the upregulation of pro-apoptotic pathway (i.e., BAD/BCL-2, BAX/BCL-XL, and caspase 3), reinforces the vulnerability of neuronal function to MeHg [81]. However, it is necessary to investigate if these pathophysiologic mechanisms described above during prenatal exposure reflects those that occur during adolescence intoxication.” (line: 276-288, page 06).

“As expected, glutamatergic neurotransmission impairment was also evidenced in an acute MeHg exposure during pregnancy (8 mg/kg, on gestational-days 8 or 15) [82]. These glutamate signaling negative effects have been related to lower level of explora-tory behavior, as well as a deficiency in habituation behavior on the new object explo-ration task [82].” (line: 289-293, page 06).

Line 230: worthy of mention is Brancato, A., Plescia, F., Lavanco, G., Cavallaro, A., Cannizzaro, C.Continuous and intermittent alcohol free-choice from pre-gestational time to lactation: Focus on drinking trajectories and maternal behavior(2016) Frontiers in Behavioral Neuroscience, 10 (MAR), art. no. 31, 11

Response: We thanks for your guidance. We included the findings of Brancato et al. 2016 work, as follow: “It is relevant to note that the severity of alcohol toxicity effects is related to drinking pattern and time of exposure, which intermittent alcohol exposure induces to increased alcohol intake during pre-gestational period and lactation than the continuous access [86].” (line: 315-318, page 07).

Lines 264-7: also acetaldehyde has risen concern in its ability to alter emotional, reactivity, learning and memory. see for reference

Plescia, F., Brancato, A., Marino, R.A.M., Vita, C., Navarra, M., Cannizzaro, C. Effect of acetaldehyde intoxication and withdrawal on NPY expression: Focus on endocannabinoidergic system involvement (2014) Frontiers in Psychiatry, 5 (OCT), art. no. 138

Cacace, S., Plescia, F., La Barbera, M., Cannizzaro, C. Evaluation of chronic alcohol self-administration by a 3-bottle choice paradigm in adult male rats. Effects on behavioural reactivity, spatial learning and reference memory (2011) Behavioural Brain Research, 219 (2), pp. 213-220.

Response: We thank the Reviewer for these suggested corrections, with which we fully agree. The indicated papers are now quoted in the revised manuscript, as follow: “Curiously, acetaldehyde, the first oxidation product of alcohol and one of the mediators of its peripheral and central effects, also has rewarding, motivational, and additive properties most likely due to its ability to affect the dopaminergic and endocannabinoid systems [89]. Such findings highlight the multiple targets affected by EtOH and its metabolites in CNS. Finally, a study where adult rats exposed to the free-access paradigm indicated that a moderate alcohol consumption was able to induce an increase in behavioral reactivity [90].” (line: 327-332, page 07).

Line 290: does this research detect direct correlation?

Response: We very much thank the Reviewer for requesting this clarification. This research did not detect a direct correlation, since was performed only epidemiological data research of symptomatologic problems mentioned to patients with mercury levels found, however deep investigations are necessary due evidence demonstrates that mercury elicits psychiatric disturbance, as anxiety and depression, as well as changes in sleep biomarkers. These suggest that mercury exposure and ethanol/mercury simultaneous exposure may be considerate a biological factors risk. To clarify such issue, we introduced in the sentence: “Although these two studies have not detected a direct correlation, such symptoms were recorded, highlighting the importance of identify sleep biomarkers in MeHg intoxication.” (line: 379-381, page 08).

Lines 298-301:  the authors may intend to deepen this paragraph??

Response: As requested, the paragraph was adjusted, as follow: “In fact, there are two process of sleep regulation called homeostatic and circadian, which the former regulates the length and depth of sleep, whereas the later regulate patterns of maximum sleepiness and maximum alertness throughout the 24h per day [96]. Some studies have claimed that such altered homeostatic sleep system induces mood disorders (increase of the emotional reactivity/impulsivity), decreasing the connectivity between prefrontal cortex and amygdala, and consequently increasing sleep deprivation, affecting numerous executive functions as attention, working memory, inhibition, and cognitive flexibility [97].” (line: 402-409, page 08).

Lines 302-7: sentences lack correlation or link between cause and effect. In this case, between neurotransmitters/ sleep/ mercury. Authors should formulate more focused sentences without digressions.

Response: We thank the Reviewer for noting this poorly explained relationship, which is now expanded and corrected in the revised manuscript. “Interactions between neurotransmitter systems that affect circadian cycle markers, such as melatonin (one of the sleep cycle promoters), can predispose to sleep disorders. In mammalians, noradrenergic inputs of suprachiasmatic nucleus (integrative part of hy-pothalamus) to pineal gland results in expression of serotonin N-acetyltransferase, an enzyme that converts serotonin in N-acetylserotonin, which is posteriorly methylated by hydroxyindole-O-methytransferase generating melatonin [99]. Such noradrenaline re-leased by suprachiasmatic nuclei to promotes melatonin synthesis during the night and the expression of serotonin N-acetyltransferase are negatively regulate by activation of the glutamatergic system [100]. Considering that mercury induces hyperactivation of glu-tamatergic pathway, it is reasonable to conclude the involvement of glutamate on sleep disturbance provoked by mercury intoxication, which deserves further investigation. Furthermore, animal studies revealed that chronic MeHg exposure elicits up-regulation of acetylcholine receptor and acetylcholinesterase inhibition [101], allowing the activation of dopamine projections involved in the cortico-striato-pallidal via, also regulated by glu-tamatergic projections, affecting the sleep-wake state [102].” (line: 410-424, page 08-09).

Line 313: sentence not entirely correct, please specify pathological conditions that lead to the exacerbated endocrine response involving HPA axis

Response: We apologize for our poorly explanation. As requested, we specify pathological conditions that lead to the exacerbated endocrine response involving HPA axis, as follow: “Although the knowledge that mercury chloride, MeHg, or mercury vapor can form deposits in the hypothalamus of rats, the impact of this metal deposit in the sleep cycle is unclear [104, 105]. An experimental study has postulated that citrate mercuric administration promotes changes in the circadian rhythm by reduction of cytoplasmic RNA of neurons in hypothalamus, reflecting in hypothalamic dysfunction [106]. In this context, several possible conditions may be involved: (I) mercury accumulation in the pineal gland decreases melatonin and serotonin secretion [107]; (II) low melatonin levels decrease protective mechanisms important in the detoxification of mercury, as well as selenium and glutathione [108]; (III) these serotonin and melatonin reductions induce adaptive responses to free radicals formed, as overall production of NO, mainly on the suprachiasmatic nucleus (integrative part of the HPA axis), which affects operate capacity of the suprachiasmatic nucleus in a similar manner to a light signal stimulus [96]; (IV) Such possible abnormalities may result in endogenous desynchronies in light/dark cycle, as cortisol levels, sleep/wakefulness, and temperature [109]. Besides, mercury can deposit on pituitary gland in animals and humans, principally by intracellular changes [110, 111]. All of these hypothesized dysfunctions may impact on sleep alterations and for this reason deserves further well-design investigations”. (line: 431-447, page 09).

We finalize the hypothesis with the sentence already present in the previous manuscript version, as follow: “Possible damage of the HPA axis or stress situation reinforce the idea of sleep disorders that result in release of these hormones, as observed in studies of single ad-ministration of MeHg in rats. One hour following subcutaneous 12 mg/kg of MeHg, elevated ACTH levels with reversible peak in the serum [112]. Moreover, different forms of mercury compounds can deposit on adrenal gland in the cortex and medulla regions, interfering with adrenal maturation and increasing corticosterone release, affecting HPA axis [113].” (line: 448-453, page 09).

We thank again the Editor and the Reviewers for their comments and helpful suggestions, which contributed to allow us improving the MS. We hope that, by addressing all the comments of the Reviewers, the revised version of the MS may prove acceptable for publication in International Journal of Molecular Sciences.

Sincerely yours,

Cristiane Maia

Reviewer 2 Report

This review article discusses findings on methylmercury and ethanol exposure's influence on emotional behavior. The review is moderate in the quality of discussion. However, the review has some significant weaknesses: For example, the addition of mercury rate and concentration exposure and its clinical and societal burden would be helpful. In addition, is there any report of prenatal mercury exposure, especially during pregnancy, and its impact on offspring development and cognitive function? It is essential to report the concentration of mercury and exposure period (acute or chronic) and the developmental period used in each study that looked at the emotional behavior. While mercury effects are well described for each behavior but ethanol and mercury effects are not well discussed. The review used only the adolescent period, so it is better to avoid the neurodevelopment term throughout the study.

Author Response

October 25, 2021

MDPI IJMS Editorial Office

Manuscript: ijms-1403512

Title: "Methylmercury plus ethanol exposure: how much this combination affects

emotionality?"

Dear Editor,

We first wish to thank you for the opportunity to allow a resubmission of a revised and now ameliorated manuscript. We extend our thanks to the Reviewers for their positive evaluation of our study and for their generous suggestions to clarify the manuscript: their time and efforts were very warmly appreciated. We addressed all their criticisms and, guided by the suggestions, we revised the manuscript accordingly. The detailed answers to each of the questions raised are tendered point-by-point, in order of appearance, as follows:

#Reviewer 2

This review article discusses findings on methylmercury and ethanol exposure's influence on emotional behavior. The review is moderate in the quality of discussion. However, the review has some significant weaknesses: For example, the addition of mercury rate and concentration exposure and its clinical and societal burden would be helpful. In addition, is there any report of prenatal mercury exposure, especially during pregnancy, and its impact on offspring development and cognitive function? It is essential to report the concentration of mercury and exposure period (acute or chronic) and the developmental period used in each study that looked at the emotional behavior. While mercury effects are well described for each behavior but ethanol and mercury effects are not well discussed. The review used only the adolescent period, so it is better to avoid the neurodevelopment term throughout the study.

Response: Dear reviewer, we agree to the very important comments and questions about our manuscript. To attend your solicitations, we made some adjustments to the text, even to clarify the objectives of this review. Our aim was to collect data about the effects of intoxication in adolescence, since data at this stage of development are scarce. Furthermore, in addition to focusing on the search in adolescence, we also only address the keywords depression, anxiety, and insomnia disorders. In view of the low availability of articles, when we found data in adults or perinatal exposure, which if they were important for the construction of the empirical hypotheses, we reported these studies. Thus, to clarify this issue, we added this information in the revised manuscript, as follow: “Posteriorly, we hypothesize whether mercury exposure associated to EtOH intake can synergically potentiate the damage caused by mercury per se, including the molecular mechanism involved in the toxicological behavior disturbance. Due to the importance of developmental alterations that undergo immature brain and few studies performed on this life stage, we have chosen to discuss the possible toxicological effects on adolescence period. In the absence of mercury plus EtOH studies, findings related to isolated substances were discussed to propose the toxicological mechanism. Furthermore, due to the gap in the literature at this stage of development, some findings of different period of life were also gathered to construct the hypothetical toxicological mechanism when opportune.” (line: 89-98, page 02).

In addition to these modifications, we have added information about the doses and period used in each study, as requested. Throughout our discussion, we have included highlights about the purpose of our review, which is focused on the effects of toxicants on emotionality disorders during adolescence.

We thank again the Editor and the Reviewers for their comments and helpful suggestions, which contributed to allow us improving the MS. We hope that, by addressing all the comments of the Reviewers, the revised version of the MS may prove acceptable for publication in International Journal of Molecular Sciences.

Sincerely yours,

Cristiane Maia

Reviewer 3 Report

Review of manuscript entilted: “Methylmercury plus ethanol exposure: how much this combination affects emotionality?” authored by Diandra Araújo Luz, Sabrina de Carvalho Cartágenes, Cinthia Cristina Sousa de Menezes da Silveira, Bruno Gonçalves Pinheiro, Kissila Márvia Matias Machado Ferraro, Luanna de Melo Pereira Fernandes, Enéas Andrade Fontes-Júnior, Cristiane do Socorro Ferraz Maia

Thank you for possibility to review this interesting manuscript.     

In the presented manuscript authors reviewed influence of exposure to methylmercury combined with ethanol on anxiety, depression and insomnia. In my opinion the manuscript is well-written, first the reader is introduced to basic knowledge about methylmercury itself and its kinetics, then authors refer to number of research (some self-citations, which show that authors performed experiments and are experts in this matter) about changes in behavior after exposure to methylmercury and ethanol. I found this manuscript very interesting, however I have some concerns listed below.

Major concerns:

  • Authors stated that “Our findings fail to indicate depressive-like behavior in offspring adult life following concomitant perinatal exposure to MeHg plus EtOH [27].”. After that authors stated that this result was unexpected. I think that is worth to mention that ethanol has anti-depressive properties and the lack of depressive-like behavior caused by MeHg may be attenuated by ethanol exposure. Recent study showed that exposure of mothers to ethanol caused lower depressive-like behavior in offspring (in particular cases). doi: 10.3390/brainsci11050622

Minor concerns:

  • I am not a native speaker but I found some mistakes in English language and style
  • Line 179 – “In fact, EtOH blockades NMDA receptor, reduces BDNF (…)”, for me this sentence suggests that EtOH is a ligand of NMDA, please try to rephrase this, maybe “EtOH-induced blockade of NMDA receptors” would be more suitable.

Author Response

October 25, 2021

MDPI IJMS Editorial Office

Manuscript: ijms-1403512

Title: "Methylmercury plus ethanol exposure: how much this combination affects

emotionality?"

Dear Editor,

We first wish to thank you for the opportunity to allow a resubmission of a revised and now ameliorated manuscript. We extend our thanks to the Reviewers for their positive evaluation of our study and for their generous suggestions to clarify the manuscript: their time and efforts were very warmly appreciated. We addressed all their criticisms and, guided by the suggestions, we revised the manuscript accordingly. The detailed answers to each of the questions raised are tendered point-by-point, in order of appearance, as follows:

#Reviewer 3

Major concerns:

Authors stated that “Our findings fail to indicate depressive-like behavior in offspring adult life following concomitant perinatal exposure to MeHg plus EtOH [27].”. After that authors stated that this result was unexpected. I think that is worth to mention that ethanol has anti-depressive properties and the lack of depressive-like behavior caused by MeHg may be attenuated by ethanol exposure. Recent study showed that exposure of mothers to ethanol caused lower depressive-like behavior in offspring (in particular cases). doi: 10.3390/brainsci11050622

Response: We agree to this contribution and include in the discussion section the reference, as follow: “Interestingly, the relationship between EtOH consume and depression-like phenotype sex-dependently have been proposed, including epigenetic factor as a predispose factor for depressive-like behavior induced by parental EtOH exposure [75]. In this sense, we highlight those emotional disorders involving EtOH consume depends on multiple factors as dose, period of consume, developmental stages, sex, etc [76, 77].” (line: 217-221; page 05).

Minor concerns:

I am not a native speaker but I found some mistakes in English language and style

Line 179 – “In fact, EtOH blockades NMDA receptor, reduces BDNF (…)”, for me this sentence suggests that EtOH is a ligand of NMDA, please try to rephrase this, maybe “EtOH-induced blockade of NMDA receptors” would be more suitable.

Response: We thank the Reviewer’s generosity for signaling these typo errors. The fact that none of the authors is a native English speaker is a main limitation to avoid stylistic and grammatical errors. We corrected the sentence kindly indicated by the Reviewer and made an effort (hopefully sufficient) to ameliorate the English in the revised manuscript, as follow: “EtOH-induced blockade of NMDA receptor reduces glutamate-NMDA pathway activation and BDNF levels; impair several neurotransmitter systems; alters mitochondrial homeostasis; induces oxidative stress and neuroinflammation; among other toxicological mechanisms [76, 77], which appears to share common damage pathways between both toxicants [21, 51, 52, 53, 54, 55, 56, 57, 58, 59, 67, 68, 69, 70].” (line: 223-227; page 05).

We thank again the Editor and the Reviewers for their comments and helpful suggestions, which contributed to allow us improving the MS. We hope that, by addressing all the comments of the Reviewers, the revised version of the MS may prove acceptable for publication in International Journal of Molecular Sciences.

Sincerely yours,

Cristiane Maia

Reviewer 4 Report

Combined exposure to methylmercury and ethanol is an important issue that can actually occur. Therefore, this manuscript is very interesting.

The authors reported that “The levels of intoxication considered accepted in humans is equivalent to 10 μg/g of total mercury content in hair,”(lines 39-40). This number is strictly similar to the Non Observed Adverse Effect Level (NOAEL), which is a bit different from the ACCEPTABLE LEVEL. At least, the uncertainty factor is not taken into account. This reviewer would like to request a strict description including citations from the literature.

Grandjean also suggested a toxicokinetic interaction between ethanol and (methyl-)mercury. (Environ Res, 1993; 61(1):176-83. doi: 10.1006/enrs.1993.1062.) This previous report hasn't been retested in humans much since then, but could the authors please add some comments to his report?

The harmful effects of methylmercury are highly dependent on the level of exposure. The exposure level used in animal experiments is not the same as the current human exposure level, but what do the authors think about this difference in exposure levels including MeHg and EtOH?

The authors reported that “Several studies postulate the close relationship between MeHg and increased risk for depression [35, 36]” (lines 110-111). The ref #35 reported only the results of meta-analysis of RCTs regarding the antidepressant efficacy of omega 3 fattyacids, and there was no data regarding the association of MeHg. The next ref #36 concluded that low-dose MeHg did not seem to be associated with depression in ABSTRUCT. Therefore, this sentence/paragraph should be reworded.  

The authors reported that “In the Arrifano et al. [5] study, such depression profile was associated to an increase of S100B mRNA levels, a biomarker of brain damage.”(lins 154-155). However, in the ref #5, (1) no differences in most symptoms were observed between two groups, the high-Hg group and the low-Hg group, and symptoms such as lack of attention, fine tremors, and fatigue/tiredness were more prevalent in the low exposure group (Table 3 in the reference), and (2) the #5 authors did not provide any data regarding the association between depression profile and S100B mRNA. Therefore this chapter should be absolutely reworded or removed.

The authors referred ref #66 as “Robust literature has identified EtOH intake as a risk factor in depression [66].” However, this reference reported the neurologic signs among methylmercury-poisoning areas in Minamata, Japan. There was no information between alcoholic drinking and depression. Perhaps, this ref #66 might be the ref #68 to present the association between Alcohol Use Disorder and depression.

Including these references, the authors are encouraged to check/reword all citations to make a review article.  

Author Response

October 25, 2021

MDPI IJMS Editorial Office

Manuscript: ijms-1403512

Title: "Methylmercury plus ethanol exposure: how much this combination affects

emotionality?"

Dear Editor,

We first wish to thank you for the opportunity to allow a resubmission of a revised and now ameliorated manuscript. We extend our thanks to the Reviewers for their positive evaluation of our study and for their generous suggestions to clarify the manuscript: their time and efforts were very warmly appreciated. We addressed all their criticisms and, guided by the suggestions, we revised the manuscript accordingly. The detailed answers to each of the questions raised are tendered point-by-point, in order of appearance, as follows:

#Reviewer 4

The authors reported that “The levels of intoxication considered accepted in humans is equivalent to 10 μg/g of total mercury content in hair,”(lines 39-40). This number is strictly similar to the Non Observed Adverse Effect Level (NOAEL), which is a bit different from the ACCEPTABLE LEVEL. At least, the uncertainty factor is not taken into account. This reviewer would like to request a strict description including citations from the literature.

Response: We thank the Reviewer for these suggested corrections, with which we fully agree. We adequate this topic according to required “Non-observed adverse effect level (NOAEL) declared by World Health Organization (WHO) in human hair mercury concentration of 10 μg g-1 has been exceeded in contaminated regions population, as in mining areas [5]. Despite WHO NOAEL recommendation, it fails to delimitate relevant factors as low levels of long-term exposure still occurring or the past acute intoxication still detected in hair [6]. Thus, the establishment of secure levels of mercury exposure is very complex, as well as the design of intoxication paradigm in animal models which could reflect human exposure. Furthermore, mercury distribution, toxicity, and metabolism depend on its chemical form, which methylmercury (MeHg), an organic derivate, has been extensively studied due its ability to cross the blood-brain barrier, reaching high levels on the central nervous system (CNS) [7].” (line: 39-49, page 01-02).

Grandjean also suggested a toxicokinetic interaction between ethanol and (methyl-)mercury. (Environ Res, 1993; 61(1):176-83. doi: 10.1006/enrs.1993.1062.) This previous report hasn't been retested in humans much since then, but could the authors please add some comments to his report?

Response: We are thankful for the suggestion and included the citation, as following: “A survey related to women that drank alcohol concomitantly to a regular consume of fishes/seafood during pregnancy support this interaction [31]. Grandjean and Weihe [31] detected low levels of mercury on cord-blood of children whose mother have consumed EtOH compared to pregnant abstinent. The authors attributed this mercury reduced level to the presence of protector compounds on fishes (i.e., polyunsaturated fatty acids), since the elevate frequency of fish by pregnant has been correlated to lower average of mercury on cord-blood [31].” (line: 111-117, page 03).

The harmful effects of methylmercury are highly dependent on the level of exposure. The exposure level used in animal experiments is not the same as the current human exposure level, but what do the authors think about this difference in exposure levels including MeHg and EtOH?

Response: We very much thank the Reviewer for requesting this clarification. This was important to re-structure the revised manuscript. To clarify such issue, we now added the paragraph: “Despite WHO NOAEL recommendation, it fails to delimitate relevant factors as low levels of long-term exposure still occurring or the past acute intoxication still detected in hair [6]. Thus, the establishment of secure levels of mercury exposure is very complex, as well as the design of intoxication paradigm in animal models which could reflect human exposure. Furthermore, mercury distribution, toxicity, and metabolism depend on its chemical form, which methylmercury (MeHg), an organic derivate, has been extensively studied due its ability to cross the blood-brain barrier, reaching high levels on the central nervous system (CNS) [7]. Therefore, in vitro and in vivo studies have been essential to elucidate the toxicological mechanisms that underlie the symptoms exhibited by humans following mercury exposure. In experimental challenges, mercury has been administrated by inhalation, orally, or intraperitonially routes, while in humans the consume of contaminated foods (i.e., fishes and seafood) reflects the most commonly intoxication profile. Nonetheless, doses that elicit mercury blood and hair levels similar to that found in clinical studies, in addition differentiation of acute and chronic exposure, consist of strategies to minimize limitations related to animal studies.” (line: 41-56, page 01-02).

The authors reported that “Several studies postulate the close relationship between MeHg and increased risk for depression [35, 36]” (lines 110-111). The ref #35 reported only the results of meta-analysis of RCTs regarding the antidepressant efficacy of omega 3 fattyacids, and there was no data regarding the association of MeHg. The next ref #36 concluded that low-dose MeHg did not seem to be associated with depression in ABSTRUCT. Therefore, this sentence/paragraph should be reworded. 

Response: We thank the Reviewer for these suggested corrections, with which we fully agree and corrected. Onishchenko et al study [38], "investigated the long-term effects of developmental exposure to methylmercury (MeHg), which the forced swimming test revealed a predisposition to depressive-like behavior in the MeHg-exposed male animals", whereas, Onishchenko et al [39] "demonstrate that developmental exposure to low levels of MeHg predisposes mice to depression phenotype and induces epigenetic suppression of BDNF gene expression in the hippocampus". The citation was modified and correctly cited: “Studies postulate the relationship between long-term effects of exposure to MeHg (0.5 mg/kg/day) and increased risk for depression [38, 39].” (line: 144-145, page 03).

The authors reported that “In the Arrifano et al. [5] study, such depression profile was associated to an increase of S100B mRNA levels, a biomarker of brain damage.”(lins 154-155). However, in the ref #5, (1) no differences in most symptoms were observed between two groups, the high-Hg group and the low-Hg group, and symptoms such as lack of attention, fine tremors, and fatigue/tiredness were more prevalent in the low exposure group (Table 3 in the reference), and (2) the #5 authors did not provide any data regarding the association between depression profile and S100B mRNA. Therefore this chapter should be absolutely reworded or removed.

Response: Thank you for this observation. We clarify the issue, as follow: “An epidemiological survey with Amazon riverine inhabitants who claimed to present depression symptoms demonstrated a total mercury level in hair at levels below of 10 μg/g [27]. In turn, Arrifano et al. [5] not found statistical differences upon depression symptoms between groups with higher (≥10 µg/g) and lower (<10 µg/g) amount of total hair mercury, which can suggest that occurrence of this psychiatric disorder was not dependent of mercury levels exposure. However, complementary studies with more participants are necessary to clarify this issue, characterizing fundamental issues, as if the exposure is acute or chronic, as a well as to establish a more concrete relationship between depressive symptoms and mercury exposure.” (line: 200-208, page 04-05).

The authors referred ref #66 as “Robust literature has identified EtOH intake as a risk factor in depression [66].” However, this reference reported the neurologic signs among methylmercury-poisoning areas in Minamata, Japan. There was no information between alcoholic drinking and depression. Perhaps, this ref #66 might be the ref #68 to present the association between Alcohol Use Disorder and depression.

Response: We thank the Reviewer for noting this omission; we provided data to support such information, as follow: “Robust literature has identified EtOH intake as a risk factor in depression [73, 74]” (line: 216, page 05).

We thank again the Editor and the Reviewers for their comments and helpful suggestions, which contributed to allow us improving the MS. We hope that, by addressing all the comments of the Reviewers, the revised version of the MS may prove acceptable for publication in International Journal of Molecular Sciences.

Sincerely yours,

Cristiane Maia

Round 2

Reviewer 1 Report

 Accept in present form

Reviewer 2 Report

The review article significantly improved, and the authors have addressed the majority of my inquiries. As a result, I have no more questions about this manuscript.